# CLEAN-NeRF: DEFOGGING USING RAY STATISTICS PRIOR IN NATURAL NeRFS

## ABSTRACT

State-of-the-art Neural Radiance Fields (NeRFs) still struggle in novel view synthesis for complex scenes, producing inconsistent geometry among multi-view observations, which is manifested into foggy "floaters" typically found hovering within the volumetric representation. This paper introduces *Clean-NeRF* to improve NeRF reconstruction quality by directly addressing the geometry inconsistency problem. Analogous to natural image statistics, we first perform empirical studies on NeRF ray profiles to derive the *natural ray statistics prior*, which is employed in our novel ray rectification transformer capable of limiting the density only to have positive values in applicable regions, typically around the first intersection between the ray and object surface. Moreover, Clean-NeRF automatically detects and models view-dependent appearances to prevent them from interfering with density estimation. Codes will be released.

## 1 INTRODUCTION

Neural Radiance Fields (NeRFs) (Mildenhall et al., 2020) have seminal contribution in the fields of novel view synthesis (Barron et al., 2021; Müller et al., 2022; Chen et al., 2022; Tancik et al., 2023), AR/VR (Zhang et al., 2021a; Li et al., 2022; Wang et al., 2022; Attal et al., 2023), digital human (Zhao et al., 2022; Işık et al., 2023; Kirschstein et al., 2023), and 3D content generation (Chan et al., 2022; Poole et al., 2023; Lin et al., 2022). To date, unfortunately, NeRF and its many variants encounter challenges when the reconstructed scene lacks view consistency and has distractors such as objects that are transitory or have specular appearances.

(Although NeRF typically conditions radiance colors on viewing directions, it can often model view-dependent effects such as specularity incorrectly due to insufficient input observations.) Such distractors in the input observations give rise to ambiguity, resulting in a "foggy" density field as shown in Fig. 1.

To address these issues, some approaches utilize semantic segmentation models to mask out distractors (Rematas et al., 2022; Tancik et al., 2022). Dynamic NeRFs (Pumarola et al., 2021; Zhang et al., 2021a; Park et al., 2021; Wu et al., 2022) can to some extent interpret distractors as dynamics within the scene. NeRF-W (Martin-Brualla et al., 2021) explains photometric and environmental variations between images by learning a shared appearance representation. (Sabour et al., 2023) and (Goli et al., 2023) estimate robustness or uncertainty and down-weight photometrically-inconsistent observations during optimization. NeRFbuster (Warburg et al., 2023) leveraged a local 3D diffusion geometric prior to encouraging plausible geometry.

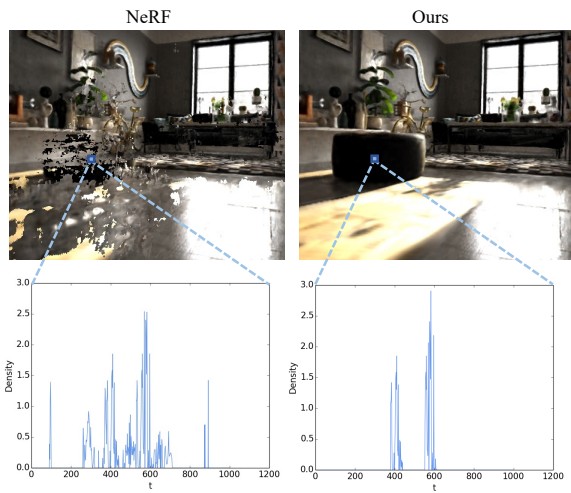

Figure 1: ***Foggy* vs *Clear* NeRF.** Our Clean-NeRF avoids foggy "floaters". Below are density profiles along a given ray from vanilla NeRF and our approach.

In contrast to previous approaches, we propose to rectify NeRF representations from the aspects of geometry and appearance. On one hand, we introduce a ray rectification transformer that utilizes an MAE-like strategy to learn priors about ray profiles, capable of eliminating unreasonable density emergence along each ray. On the other hand, we extend the vanilla NeRF radiance color estimation module to account for view-inconsistent observations. In this way, we manage to disentangle view-dependent and view-independent appearances.

Experiments verify that our proposed Clean-NeRF can effectively get rid of floater artifacts. In summary, our contributions include the following:

- We propose a ray rectification transformer with an MAE-like training strategy to learn ray profiles and eliminate incorrect densities along rays.
- We disentangle view-independent and view-dependent appearances during NeRF training to eliminate the interference caused by view-inconsistent observations.
- Extensive experiments and ablations verify the effectiveness of our core designs and results in improvements over the vanilla NeRF and other state-of-the-art alternatives.

## 2 RELATED WORKS

**Neural Representations.** Recent advancements in coordinate-based neural representations, also known as neural fields, have significantly propelled the neural processing capabilities for 3D data and multi-view 2D images (Sitzmann et al., 2019; Mescheder et al., 2019; Park et al., 2019; Mildenhall et al., 2020). The seminal work NeRFs debut in (Mildenhall et al., 2020) has achieved unprecedented effects in novel view synthesis by modeling the underlying 3D scene as a continuous volumetric field of color and density using multi-layer perceptrons (MLPs). Subsequent research has broadened the capabilities of NeRF models in various respects, encompassing the acceleration of training and inference (Yu et al., 2021; Fridovich-Keil et al., 2022; Chen et al., 2022; Müller et al., 2022; Chen et al., 2023), the modeling of dynamic scenes (Zhang et al., 2021a; Park et al., 2021; Pumarola et al., 2021; Tretschk et al., 2021), and the improvement of scene understanding (Zhi et al., 2021; Kobayashi et al., 2022; Liu et al., 2022; Fan et al., 2023; Kerr et al., 2023). the relaxation of stringent camera calibration requirements (Lin et al., 2021; Meng et al., 2021; Bian et al., 2023), Despite tremendous progress, NeRFs demand hundreds of input images and fall short in synthesizing novel views under conditions of sparse observations, curtailing their prospective applications in real-world scenarios.

**Reflectance Decomposition.** To acquire reflectance data, sophisticated devices have traditionally been necessary to sample the light-view space (Kang et al., 2018; Matusik et al., 2003; Nielsen et al., 2015). Subsequent research has proposed practical techniques for acquiring spatially varying BRDFs, such as those presented in (Kang et al., 2018; Matusik et al., 2003; Nielsen et al., 2015; Nam et al., 2018). More recently, deep learning methods have made it possible to acquire BRDF information from a single flash image (Li et al., 2018b;a; Deschaintre et al., 2018).

In the context of NeRF, highly reflective objects can pose challenges in the reconstruction and relighting process. Previous works have attempted to address this issue by decomposing appearance into scene lighting and materials, but these methods assume known lighting (Bi et al., 2020; Srinivasan et al., 2021) or no self-occlusion (Boss et al., 2021a;b; Zhang et al., 2021b). Ref-NeRF (Verbin et al., 2022) uses a representation of reflected radiance and structures this function using a collection of spatially-varying scene properties to reproduce the appearance of glossy surfaces. Despite these advances, Ref-NeRF requires accurate normal vectors and outgoing radiance estimation, which is difficult to obtain for sparse input views. In addition, effectively addressing the view-dependent appearance problem in the context of large scenes and sparse observations remains a challenge. To address this issue, we propose a simple yet effective decomposition method to eliminate its interference without the need to estimate surface normals or outgoing radiance.

**Intrinsic Image Decomposition.** Barrow and Tenenbaum introduced intrinsic images as a valuable intermediate representation for scenes (Barrow et al., 1978), assuming that an image can be expressed as the pointwise product of the object's true colors or reflectance and the shading on that object. This can be represented as $I = R \cdot S$, where $I$, $R$, and $S$ denote the image, the reflectance, and the shading, respectively.

Early optimization-based works addressed the problem of separating an image into its reflectance and illumination components by assuming that large image gradients correspond to reflectance changes and small gradients to lighting changes (Land & McCann, 1971; Horn, 1974). Incorporation of additional priors improves the accuracy and robustness, such as reflectance sparsity (Rother et al., 2011; Shen & Yeo, 2011), low-rank reflectance (Adrien et al., 2009) and distribution difference in gradient domain (Bi et al., 2015; Li & Brown, 2014). Deep learning methods (Fan et al., 2018; Yu & Smith, 2019; Zhu et al., 2022; Li & Snavely, 2018a;b) have emerged to perform intrinsic image decomposition, estimating the reflectance and shading on labeled training data. Notably and differently, in intrinsic image decomposition, where shadows and highlights are separated as high-frequency components, these components *may* still be separated as view-independent in our Clean-NeRF as long as they are *consistent* across all input views, e.g., a static shadow is consistently observed across all views. Thus, intrinsic image decomposition is inappropriate (both overkill and inadequate) to the "vi-vd" decomposition of Clean-NeRF. IntrinsicNeRF (Ye et al., 2023) introduces intrinsic decomposition to the NeRF-based neural rendering method, which allows for editable novel view synthesis in room-scale scenes. Compared with our simple and effective appearance decomposition, IntrinsicNeRF requires dense inputs (900 images for their indoor Replica scene), which assumes the NeRF reconstruction is accurate.

## 3 METHOD

In this section, we propose Clean-NeRF to address the floater artifacts caused by view-inconsistent observations effectively. In Sec. 3.1, we introduce the overall architecture of Clean-NeRF compared to the vanilla NeRF. Sec. 3.2 describes our ray rectification transformer with an MAE-like training strategy to learn ray profiles and eliminate incorrect densities along rays. Sec. 3.3 describes our principled implementation for decomposing an appearance into view-dependent ("vd") and view-independent ("vi") components.

### 3.1 OVERALL ARCHITECTURE

Fig. 2 compares vanilla NeRF (Mildenhall et al., 2020) and our model, both taking a sampled spatial coordinate point $\mathbf{x} = (x, y, z)$ and direction $\mathbf{d} = (\theta, \phi)$ as input and output the volume density and color. Vanilla NeRF uses a spatial MLP to estimate volume density $\sigma$ at position $\mathbf{x}$. Then the directional MLP takes as input the direction $\mathbf{d}$ as well as spatial feature $\mathbf{b}$ to estimate a view-dependent output color $\mathbf{c}$. In Clean-NeRF architecture, we also use the spatial MLP and the directional MLP

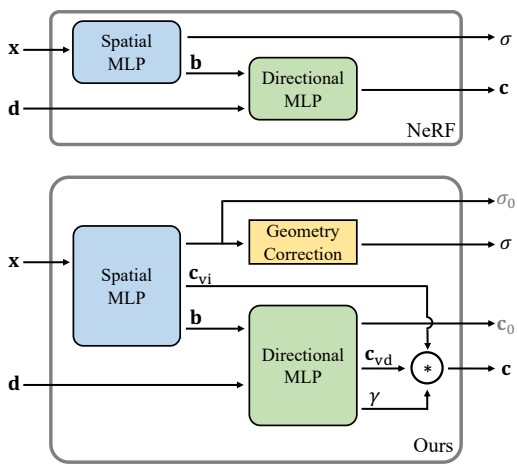

Figure 2: **Vanilla NeRF and Clean-NeRF architectures.** We propose a "vi-vd" appearance decomposition based on spherical harmonics (SHs) to account to view-inconsistent appearances, with a ray rectification transformer to refine ray profiles.

to output an initial estimation of density $\sigma_0$ and color $\mathbf{c}_0$, similar to the vanilla NeRF. The initial estimation of color corresponding to ray $\mathbf{r}(t) = \mathbf{o} + t\mathbf{d}$ can be evaluated from $\sigma_0$ and $\mathbf{c}_0$:

$$\hat{\mathbf{C}}_0 = \sum_{k=1}^{K} \hat{T}_0(t_k)\alpha(\sigma_0(t_k)\delta_k)\mathbf{c}_0(t_k), \tag{1}$$

where $\hat{T}_0(t_k) = \exp\left(-\sum_{k'=1}^{k-1} \sigma_0(t_k)\delta(t_k)\right)$, $\alpha(x) = 1 - \exp(-x)$, and $\delta_p = t_{k+1} - t_k$. Without requiring additional inputs, Clean-NeRF differs from vanilla NeRF in that while making the initial estimation, we also predict the view-independent color component $\mathbf{c}_{vi}$ with the spatial MLP and the

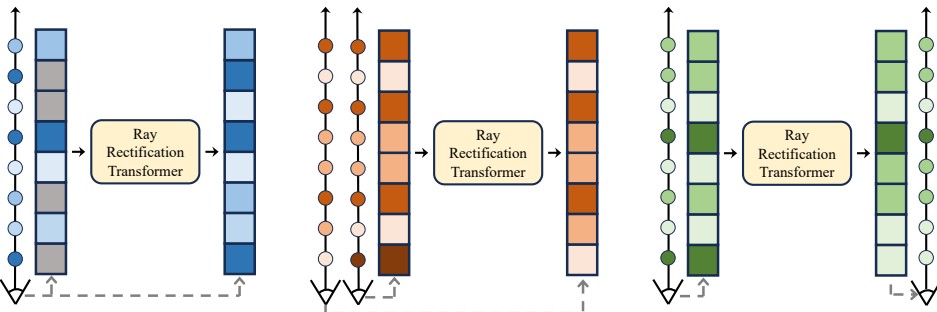

Figure 3: **1)** We use an MAE-like training method to learn the prior about ray profiles. **2)** We use paired coarse-fine ray profiles to fine-tune the network. **3)** During NeRF training, we use a ray rectification transformer to correct the density distribution of the rays.

view-dependent color component $\mathbf{c}_{\text{vd}}$ with the directional MLP. We supervise these estimations by performing an SH-based decomposition of the initial color estimation $\mathbf{c}_0$, as described in Sec. 3.3. The directional MLP also estimates a view-dependent factor $\gamma$ and we obtain a final color estimation $\mathbf{c}$ by:

$$\mathbf{c} = \gamma\mathbf{c}_{\text{vi}} + (1 - \gamma)\mathbf{c}_{\text{vd}}. \tag{2}$$

Note that $\mathbf{c}_{\text{vi}}$ captures the overall scene color while $\mathbf{c}_{\text{vd}}$ captures the color variations due to changes in viewing angle. By blending these two components with the factor $\gamma$, we can synthesize a faithful color of the underlying 3D scene, even from a limited number of input views. To better eliminate the floating artifacts, we propose a ray correction transformer to correct the initial density estimation and obtain the final density estimation $\sigma$, as described in Sec. 3.2. The final estimation of color corresponding to ray $\mathbf{r}(t) = \mathbf{o} + t\mathbf{d}$ is then computed as

$$\hat{\mathbf{C}} = \sum_{k=1}^{K} \hat{T}(t_k)\alpha(\sigma(t_k)\delta_k)\mathbf{c}(t_k), \tag{3}$$

where $\hat{T}(t_k) = \exp\left(-\sum_{k'=1}^{k-1} \sigma(t_k)\delta(t_k)\right)$, $\alpha(x) = 1 - \exp(-x)$, and $\delta_p = t_{k+1} - t_k$. We train the network using photometric loss based on both the initial and the final estimations

$$\mathcal{L}_{\text{pho}} = \sum_{\mathbf{r} \in \mathcal{R}} \left( \left\| \hat{\mathbf{C}}_0(\mathbf{r}) - \mathbf{C}(\mathbf{r}) \right\|_2^2 + \left\| \hat{\mathbf{C}}(\mathbf{r}) - \mathbf{C}(\mathbf{r}) \right\|_2^2 \right). \tag{4}$$

## 3.2 RAY RECTIFICATION TRANSFORMER

Clean-NeRF is consistent with standard NeRF in that an initial density estimation, denoted as $\sigma_0$, is generated for volume rendering with the initial color estimation $\mathbf{c}_0$. We propose a geometry correction strategy for the final rendering that simultaneously refines the density estimation while better handling unsightly floater artifacts.

**Density priors through masked autoencoders.** Masked autoencoders (MAE) have shown themselves as scalable self-supervised learners. Inspired by their success, we learn a transformer model $F_\Theta$ to capture NeRF ray geometry priors, where $\Theta$ denotes learnable parameters. Specifically, we first perform NeRF reconstruction on numerous scenes, each with sufficient input observations. This process generates a set of ray profiles $\{\sigma_i\}$ for training our model. To train a geometry correction module $F_\Theta$, we first perform MAE pre-training. Specifically, the training loss is:

$$\mathcal{L}_{\text{MAE}} = \sum_{i=1}^{N} \|F_\Theta(\mathbf{m}_i\sigma_i) - \sigma_i\|_2^2 \tag{5}$$

where $\mathbf{m}$ is a random binary mask that hides a portion of the density values, thus requiring the model to predict them.

**Ray rectification Transformer** With self-supervised pretraining, we generate paired coarse and fine rays to fine-tune this model, enabling it to predict fine rays from coarse rays. Specifically, we train two versions of NeRF for the same scene: a fine version of NeRF trained on all images, and a coarse version of NeRF trained on only one-fifth of the images. By rendering from the same viewpoint, these two opposing NeRFs can generate paired training data samples $\{\sigma_{i,\text{coarse}}, \sigma_{i,\text{fine}}\}$. We then fine-tune this model using:

$$\mathcal{L}_{\text{corr}} = \sum_{i=1}^{M} \|F_{\Theta}(\sigma_{i,\text{coarse}}) - \sigma_{i,\text{fine}}\|_2^2 \tag{6}$$

## 3.3 Appearance Decomposition

To guide our vi-vd decomposition, we utilize Spherical harmonics (SHs) which are widely used as a low-dimensional representation for spherical functions, and have been used to model Lambertian surfaces (Ramamoorthi & Hanrahan, 2001; Basri & Jacobs, 2003) as well as glossy surfaces (Sloan et al., 2002). To use SH functions to model a given function, we query the SH functions $Y_\ell^m : \mathbb{S}^2 \mapsto \mathbb{R}$ at a viewing angle $\mathbf{d}$ and then fit the estimation $\mathbf{c}_0$ by finding the corresponding coefficients. We use low-degree SH functions to compute ideal values of view-independent color components, and high-degree SH functions for view-dependent components. In this subsection, we will perform all of our calculations at an arbitrary position $\mathbf{x}$ in space, and therefore we will omit the symbol $\mathbf{x}$ from our notation.

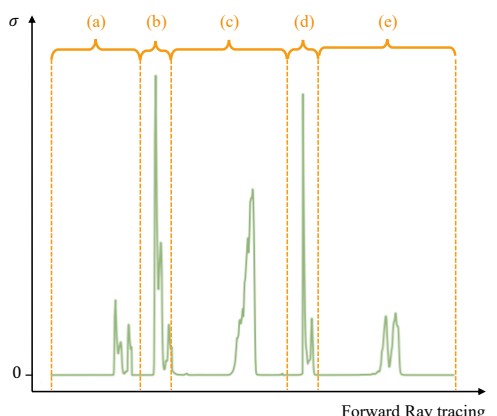

Figure 4: **An example traced ray's density profile.** Peaks in region (a) and region (e) appear as floating artifacts and would interfere with the rendering process.

We use $\mathbf{y}(\mathbf{d}) \in \mathbb{R}^L$ to represent the set of SH function values at the viewing angle $\mathbf{d}$:

$$\mathbf{y}(\mathbf{d}) = \left[Y_0^0(\mathbf{d}), Y_1^{-1}(\mathbf{d}), Y_1^0(\mathbf{d}), Y_1^1(\mathbf{d}), \dots, Y_{\ell_{\max}}^{\ell_{\max}}(\mathbf{d})\right]^\top, \tag{7}$$

where $L = (\ell_{\max} + 1)^2$. To ensure clarity, we will use $c : \mathbb{S}^2 \mapsto \mathbb{R}$ to represent one of the three channels of $\mathbf{c}_0$ at a given position $\mathbf{x}$ (also $c_{\text{vi}}$ and $c_{\text{vd}}$), noting the derivation should be readily extended to all three channels. We begin by sampling a set of $N$ viewing angles $\mathbf{d}_i, 1 \leq i \leq N \subset \mathbb{S}^2$. The colors of all the sample directions are represented using a vector $\mathbf{s} \in \mathbb{R}^N$:

$$\mathbf{s} = \begin{bmatrix} c(\mathbf{d}_1) & c(\mathbf{d}_2) & \dots & c(\mathbf{d}_{N-1}) & c(\mathbf{d}_N) \end{bmatrix}^\top \tag{8}$$

The coefficients to be determined are represented by a vector $\mathbf{k} \in \mathbb{R}^L$. To find the optimal coefficients that fit the view-dependent color estimation, we solve the following optimization problem:

$$\min_{\mathbf{k} \in \mathbb{R}^L} \|\mathbf{s} - \mathbf{Y}\mathbf{k}\|_2^2, \tag{9}$$

where

$$\mathbf{Y} = \begin{bmatrix} \mathbf{y}(\mathbf{d}_1) & \mathbf{y}(\mathbf{d}_2) & \dots & \mathbf{y}(\mathbf{d}_{N-1}) & \mathbf{y}(\mathbf{d}_N) \end{bmatrix}. \tag{10}$$

This is a standard linear regression problem, where we seek to find the values of the coefficient vector $\mathbf{k}$ which minimizes the least squares error between the vector $\mathbf{s}$ and the linear combination of the columns of $\mathbf{Y}$, weighted by the coefficients in $\mathbf{k}$. Using the normal equation, the solution is given by:

$$\mathbf{k}^* = (\mathbf{Y}^\top \mathbf{Y})^{-1} \mathbf{Y}^\top \mathbf{s} \tag{11}$$

We can use the solution coefficients $\mathbf{k}^*$ as weights to linearly combine SH functions. Retaining the low-degree SH functions allows us to capture the view-independent appearance of the scene.

Conversely, including high-degree SH functions leads to a high-frequency view-dependent residue. To differentiate between the two, we denote the low-degree and high-degree functions as $L_{\text{low}}$ and $L_{\text{high}}$, respectively. To compute ideal values for the view-independent component, we can use the solution coefficients and apply the following equation:

$$\tilde{c}_{\text{vi}} = \sum_{i=1}^{L_{\text{low}}} \frac{1}{4\pi r^2} \iint_{\mathbb{S}^2} k_i^* y_i(\mathbf{d}) \sin\theta d\theta d\phi, \tag{12}$$

where we take the mean value around the $\mathbb{S}^2$ surface. In our implementation, we approximate it by

$$\tilde{c}_{\text{vi}} = \sum_{i=1}^{L_{\text{low}}} \sum_{i=1}^{N} k_i^* y_i(\mathbf{d_i}), \tag{13}$$

This value is then used to guide the output of the view-independent color component $\mathbf{c}_{\text{vi}}$ from the spatial MLP using a regularizer. Specifically, we use the following equation to compute the vi-regularizer loss $\mathcal{L}_{\text{vi}}$:

$$\mathcal{L}_{\text{vi}} = (\mathbf{c}_{\text{vi}} - \tilde{\mathbf{c}}_{\text{vi}})^2. \tag{14}$$

We apply the following equation to compute optimal values for the view-dependent component:

$$\tilde{c}_{\text{vd}}(\mathbf{d}) = \sum_{i=L_{\text{high}}}^{L} k_i^* y_i(\mathbf{d}). \tag{15}$$

Incorporating the computed value to guide the output of the view-dependent color residue $\mathbf{c}_{\text{vd}}$ from the directional MLP using the vd-regularizer loss:

$$\mathcal{L}_{\text{vd}} = \left\| \begin{bmatrix} \mathbf{c}_{\text{vd}}(\mathbf{d_1}) \\ \vdots \\ \mathbf{c}_{\text{vd}}(\mathbf{d_N}) \end{bmatrix} - \begin{bmatrix} \tilde{\mathbf{c}}_{\text{vd}}(\mathbf{d_1}) \\ \vdots \\ \tilde{\mathbf{c}}_{\text{vd}}(\mathbf{d_N}) \end{bmatrix} \right\|_2^2. \tag{16}$$

As aforementioned we consider a given position $\mathbf{x}$ in the space, while in actual implementation we take the $\ell_2$-norm among all the positions in a sampled batch for Eqn. 14 and Eqn. 16.

Refer to Fig. 4 again: peaks in region (a) and region (e) appear as floating artifacts and would interfere with the rendering process, and so they are discarded. Notably, multiple salient peaks may exist corresponding to other surface points along the ray, such as region (d), e.g., corresponding to the two slices of toast intersected by the pertinent ray in the previous figure. If the salient peak (d) is further from the ray origin but lower than (b), as Fig. 4 shows, the corresponding surface point is occluded by (b), which can be safely detected by backward pass while occlusion-correct rendering is unaffected, as peak (d) is lower. Otherwise, suppose the peak (d) is higher (e.g., the toast further back is higher), then the peak in (b) should have been clamped to reveal the true geometry (d) as the first salient peak as seen from the ray origin.

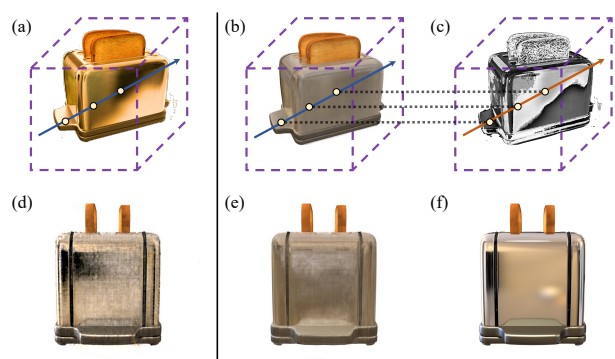

Figure 5: **Appearance Decomposition.** **(a, d)** For object surfaces with strong view-dependent effects, vanilla NeRF often gets disrupted, resulting in poor reconstruction quality. We propose decomposing appearance into **(b)** view-independent and **(c)** view-dependent components during NeRF training and incorporating geometry-related priors to improve reconstruction quality. Our approach shows the ability to **(e)** render the view-independent component of the scene and **(f)** more accurate recovery of the scene.

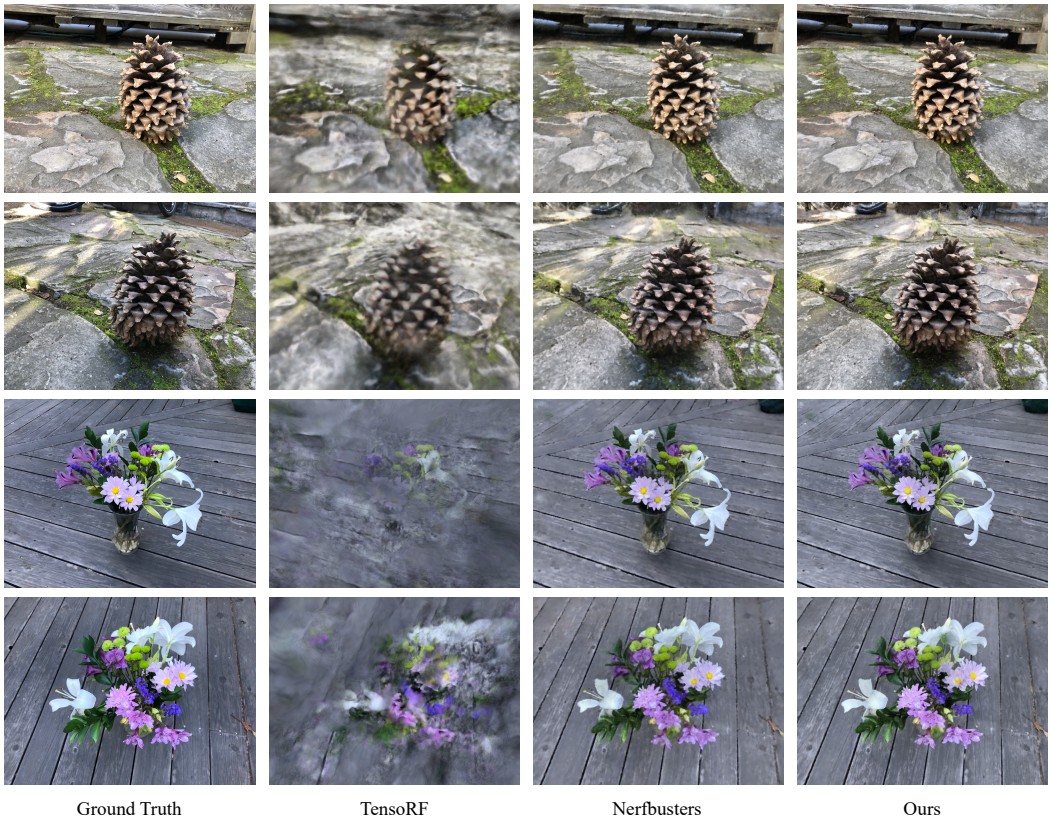

| Ground Truth | TensoRF | Nerfbusters | Ours |

Figure 6: **Qualitative evaluation of Clean-NeRF on Shiny Blender dataset.** We render the view-independent component image, and the final color image combining both view-independent and view-dependent components to compare with the ground truth.

| Method | PSNR(↑) | SSIM(↑) | LPIPS(↓) |
|---|---|---|---|
| NeRF | 15.19 | 0.475 | 0.592 |
| TensoRF | 16.35 | 0.571 | 0.533 |
| Nerfbuster | 20.94 | 0.596 | 0.406 |
| Ours | **21.05** | **0.642** | **0.313** |

Table 1: **Quantitative comparison.** We compare our proposed Clean-NeRF with representative NeRF-based methods. best second-best

## 4 RESULTS

In this section, we provide comparisons with previous state-of-the-art NeRF-based methods and evaluation of our main technical components, both qualitatively and quantitatively. We run our proposed Clean-NeRF on NeRF 360 (Mildenhall et al., 2020), a challenging real dataset of photo-realistic outdoor scenes, and compare our approach with other methods in the same field. We use the given accurate camera poses from the dataset. We train our Clean-NeRF for 500K iterations to guarantee convergence on a single NVIDIA GeForce RTX 3090 Ti GPU. All the shown cases and reported metrics are from held-out views. We report three error metrics, including peak signal-to-noise ratio (PSNR), structural similarity index measure (SSIM) (Wang et al., 2004), mean absolute error (MAE), and learned perceptual image patch similarity (LPIPS) (Zhang et al., 2018).

**Comparison on Challenging Outdoor Scenes** We compare Clean-NeRF with TensoRF (Chen et al., 2022) and Nerfbuster (Warburg et al., 2023), which are representative NeRF-based methods and strong baselines for large-scale scenes. As shown in Fig. 6 and Tab 1, our method recovers intricate details of objects in the outdoor scene.

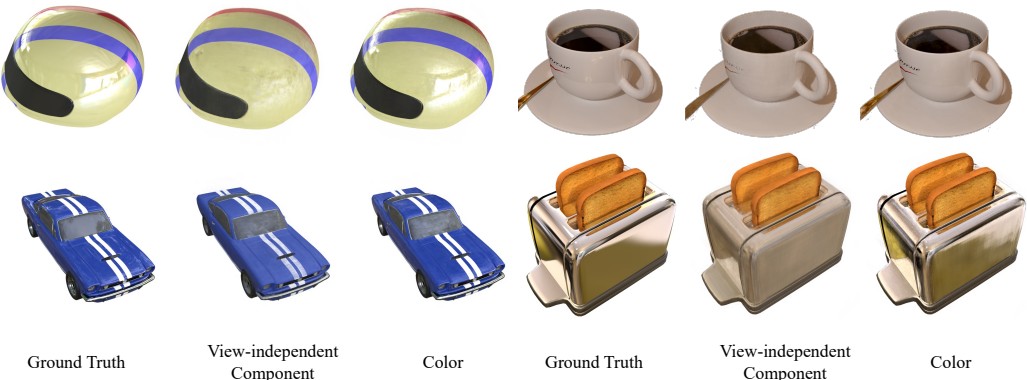

Figure 7: **Qualitative evaluation of Clean-NeRF on Shiny Blender dataset.** We render the view-independent component image, and the final color image combining both view-independent and view-dependent components to compare with the ground truth.

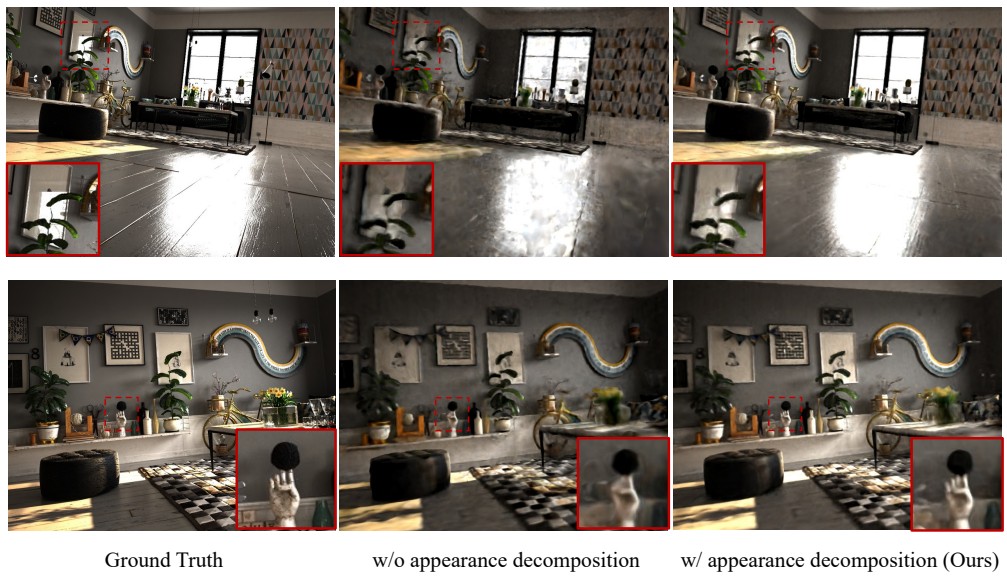

Figure 8: **Qualitative evaluation on appearance decomposition.** Without appearance decomposition, our model fails to recover the glossy objects such as the glass, the floor, and the plant.

**Appearance Decomposition on Glossy Objects** To verify Clean-NeRF's ability to decompose object appearances into the corresponding view-independent and view-dependent components, we evaluate Clean-NeRF on and render the view-independent component image and color image with both view-independent and view-dependent components (Fig. 7). The Shiny Blender dataset contains different glossy objects rendered in Blender under conditions similar to typical NeRF datasets, to verify our model's ability to handle challenging material properties by proper decomposition of observed appearance into the corresponding view-independent and view-dependent components.

**Ablation Study on our Architecture Design** We qualitatively and quantitatively evaluate the main components of Clean-NeRF on Hypersim, a Fig. 8 shows that without appearance decomposition, NeRF struggles to recover the glossy floor and plants.

## 5 DISCUSSION AND LIMITATIONS

Our method assumes fixed lighting conditions and no semi-transparent objects in a scene. In addition, we observe that, when we deal with sparse inputs and the specular highlights of a point

appear in most of the inputs, such highlights may be regarded as view-independent colors, since our method does not make any assumption about the surface properties and colors. Below, we discuss some important questions related to our work:

*Why are there floaters in sparse but not in dense inputs?* In vanilla NeRF, observation errors are backpropagated according to Eqn. 1, which are backpropagated equally to density and color along a given ray without any prior. With dense inputs, the strong geometry constraint from other view points can correct the density errors along a ray, and thus the view dependent observations will be correctly backpropagated to the color component. In contrast, when the number of inputs is limited, the network cannot resolve the ambiguity that the view dependent observations are caused by change of colors, or by the semi-transparent occluders, i.e., floaters. Since errors are backpropagated equally to both density and color along a ray, generating *floaters* is more preferable by Eqn. 1.

*What are the benefits of vi- and vd- color decomposition?* Such decomposition can stabilize the solution by reducing the ambiguity in handling view-dependent observations as residual errors in $c_{vd}$, while keeping the $c_{vi}$ stable across multiple views, thus leading to a reconstruction of higher quality. Additionally, in downstream tasks such as NeRF object detection (Hu et al., 2023) and segmentation (Ren et al., 2022), one may have to estimate the color of voxel features that is independent of viewpoints. Our $c_{vi}$ can provide such voxel feature extraction for free without additional computations.

*Why is the decomposition in Eqn. 2 correct?* Eqn. 2 can be considered as a simplified BRDF model, e.g., a simplified Phong model with diffuse and specular components but without normal and light. Although not entirely physically correct, this formulation can handle most view-dependent observations in the real world without resorting to estimating surface normals and incoming lighting conditions, thus providing a fast and easy way to optimize. According to our experiments, this formulation is generally applicable, and the resulting decomposition is reasonably accurate.

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
