# OpenReview forum: "Clean-NeRF:  Defogging using Ray Statistics Prior in Natural NeRFs"
_ICLR.cc/2024/Conference — ICLR 2024 Conference Withdrawn Submission_

### Official Review · Reviewer_wuek · 2023-10-25

**Soundness:** 2 fair
**Presentation:** 2 fair
**Contribution:** 2 fair
**Rating:** 3
**Confidence:** 4

**Summary:**

This paper proposes a novel approach to improve the quality of Neural Radiance Fields (NeRFs) by addressing the floaters problem. The paper introduces a method called Clean-NeRF, which derives natural ray statistics prior and employs a ray rectification transformer to limit density only to have positive values in applicable regions, preventing view-dependent appearances from interfering with density estimation.

The paper provides a detailed explanation of the proposed method, including the mathematical formulation of the ray rectification transformer and the natural ray statistics prior. The ray rectification transformer rectifies the density field by limiting it only to have positive values in applicable regions, typically around the first intersection between the ray and object surface. The natural ray statistics prior is derived from empirical studies on NeRF ray profiles and is used to regularize the density field during training.

The paper also presents experimental results that demonstrate the effectiveness of the proposed method in improving the quality of NeRFs. The experiments show that Clean-NeRF can produce more accurate and consistent reconstructions than state-of-the-art NeRF methods, especially in complex scenes with multiple objects and occlusions.

**Strengths:**

(1) The idea of decomposing color into view-dependent color and view-independent color is awesome. Experiments also show the effectiveness of this decomposition.
(2) The framework sounds new, e.g. using MAE to correct density estimation in NeRF.
(3) This paper provides a useful discussion at the end of the paper.

**Weaknesses:**

Weaknesses are listed below:
(1) Related works are not well discussed. Though the paper mentioned several related works, e.g. NeRFBuster (ICCV 2023), Robust-NeRF (CVPR 2023), Bayes' Rays (arXiv 2023), the technique contribution of these works are not well discussed in enough detail.

(2) This paper did not provide enough comparisons to counterparts. For example, Robust-NeRF is not compared.

(3) This paper only did limited experiments on an outdoor dataset. Why was the method not tested on the NeRFBuster dataset or the dataset provided by Robust-NeRF?

(4) On page 6 of the main paper, "Refer to Fig. 4 again...", however, the paper did not refer to Fig. 4 before page 6. Moreover, the paper did not give a clear description in the last paragraph of Page 6 of how they leverage statistics to remove floaters. For example, it is necessary to have a figure to show " the corresponding surface point is occluded by (b),".

(5) No training details are provided for the MAE. For example, how many scenes and what kind of datasets are reconstructed to train the MAE? Moreover, MAE's training time and inference time should also be provided.

(6) No ablation studies show the effectiveness of the ray rectification transformer.

(7) The paper is not well written, especially the organization of how the method works.

Overall, I think the paper is unprepared for submission to a top-tier conference. The problem this paper trying to solve is interesting and important to novel view synthesis. I have tried my best to find the strengths of the paper, I admit the idea sounds interesting to me, but I cannot convince myself to accept this paper and to give positive feedback for the current version of the paper.

**Questions:**

Questions are also included in the weaknesses.
My main concern is the generalizability and applicability of the method in tremendous real-world scenes since it only shows results in quite limited scenes.

**Details Of Ethics Concerns:**

No concerns.

---

### Official Review · Reviewer_cFcm · 2023-10-30

**Soundness:** 2 fair
**Presentation:** 3 good
**Contribution:** 2 fair
**Rating:** 3
**Confidence:** 4

**Summary:**

This paper propose Clean-NeRF to addresses the issue of inconsistent geometry and foggy "floaters" in Neural Radiance Fields (NeRFs) when dealing with complex scenes and novel view synthesis.
Clean-NeRF architecture consists of two main components: a ray rectification transformer and an appearance decomposition module. The ray rectification transformer uses an MAE-like training strategy to learn ray profiles and eliminate incorrect densities along rays.
The appearance decomposition module disentangles view-independent and view-dependent appearances during NeRF training to eliminate the interference caused by view-inconsistent observations.

**Strengths:**

The semantics of the paper is smooth, and the proposed method is simple yet effective. It achieves better performance than previous methods that have a more sophisticated procedure.

The overall framework of Clean-NeRF is very straightforward, so there should be no difficulty for others to reproduce.

The idea of decompose view-dependent color and view independent color with vd-regularizer loss is novel and interesting, it havepotential to benefit future work in this fields especially NeRF for inverse-rendering and relighting.

**Weaknesses:**

The contribution is marginal, for contribution 2, there are already a bunch of works that decompose view-dependent effect and view-independent effect also many previous works have proposed to decompose rendering into diffuse color and specular color which basically have the same meaning as vi-vd decomposition:

1) Ref-NeRF: Structured View-Dependent Appearance for Neural Radiance Fields
2) Baking Neural Radiance Fields for Real-Time View Synthesis
3) DE-NeRF: DEcoupled Neural Radiance Fields for View-Consistent Appearance Editing and High-Frequency Environmental Relighting
4) ENVIDR: Implicit Differentiable Renderer with Neural Environment Lighting
5) Neural-PIL: Neural Pre-Integrated Lighting for Reflectance Decomposition

The authors should make further clarification about how their method differs form those previous works.

Moreover, in contribution 3, the author claims their work has surpassed previous state-of-the-art methods, but the experiment part fails to support this claim. The only quantitative comparison is Table 1 which compares Clean-NeRF with vanilla NeRF, TensorRF, and Nerfbusters in *NeRF 360 dataset*.  This is not a good experiment setting, TensoRF is not designed for unbound 360 scenes, and NeRFbuster focuses on casully captured scenes.

For  *NeRF 360 dataset* the author should compare their method with that method that are designed for this task such as mip-nerf 360, MeRF. Moreover, there should also be a quantitative comparison on the Shiny Blender Dataset rather than only showing some rendering results, the author should at least compare with Ref-NeRF, which is the proposer of this dataset.

How long did it take to train Clean-NeRF? and what FPS can it achieve? The author only mentioned that they trained Clean-NeRF for 500K iterations, but didn't provide any numerical results on training time and rendering speed. adopting a transformer for ray marching have a high chance to slow down both training and inference drastically. And also made it difficult to generalize clean-nerf to those fast nerf such as instant-ngp without hurting the performance.

**Questions:**

The author claims to have conduct quantitative evaluation for ablation study. in "Ablation Study on our Architecture Design", but I can't find any quantitative result about it.

In Figure 6's caption  I believe "Shiny Blender dataset" should be "NeRF 360 dataset"

---

### Official Review · Reviewer_8rvn · 2023-11-01

**Soundness:** 1 poor
**Presentation:** 2 fair
**Contribution:** 2 fair
**Rating:** 3
**Confidence:** 4

**Summary:**

This paper proposes two contributions. The first is a "denoising" process for ray density using a self-supervised (trained with a masking technique). It takes as input the, possibly degraded by potential floating artifacts due to limited training like in a few view scenario, predicted by the NeRF to produce a cleaner version that should not be contaminated by artifacts. The second part is an appearance decomposition into a view-independent appearance contribution and a view-dependent part. This is inspired by computer graphics with simple rendering models using with Lambertian and specular reflections. The impact is shown with the  NeRF 360 dataset and a few examples of the Shiny Blender dataset.

**Strengths:**

The author proposed an interesting approach to the floaters problem. This approach deserves an in-depth study to explore its strength and limitations.

**Weaknesses:**

My first problem is with the title (and the following references). Defogging is often used in computer photography to describe the process of removing the foggy appearance to an image. This concept was then reused in NeRF to signify robustness to foggy scene or being able to "defog" the scene after having learned the foggy version. It is not clear either why the authors talk about "fog" since floaters don't follow a regular transparent density like one would expect from actual fog. Instead it's more floating random solid shapes.

The related work study is also quite poor given the setting. The authors mention in the conclusion that floaters appear in the a sparse view setting but never discuss about any work on that. This surprising since it is a very dynamic topic and, additionally, focuses on floaters. The authors can get familiar with the topic by looking at works such that {1,2,3}. This part is extremely important since it means that the experimental comparison is incomplete because it doesn't compare to the appropriate methods.

My main concern with this paper is that there is two seemingly independent contributions, the "defogging" and the appearance decomposition. It is not clear which contribution contribute to what and as such looks like two separate papers combined into one. in my opinion, it would have been better to focus on a single contribution (the "defogging"part is the most interesting in my opinion) but with a more indepth analysis.

As mentioned previously, the analysis is incomplete. For example, state-of-the-art NeRFs rely on a two sampling strategy (a coarse and a fine that depends on the output of the coarse density), how would the two sampling strategy behaves with the introduction of the transformers? My first intuition was that the transformer would be applied only after the training (as a "defogging" process of the learned scene). Additional study on the importance of the different parts of Eq. 4 would have been interesting (transformer only at the end, trained with only the $\hat{C}$ term or the two terms as proposed). It is also not entirely clear either whether the transformer is fixed during the training for a specific scene or it is also allowed to evolve (since $\hat{C)$ depends on it). I'm also surprised that no scene prior are used for the transformer. As presented, the transformer takes a single ray to perform the cleaning step. In my opinion, this is a very difficult problem since it could be possible to create scenes with any density distribution. This means that the transformer is highly dependent on the scene it was trained on and as such expect poor generalization to different and more complex scenes. I would have expected a dependence on additional information such as input images and/or neighboring rays.

The experimental section is extremely poor. First of all, the comparison is only made in a very specific setting of the 360 dataset, i.e. with a main object in the middle of the scene and a background. It would have been interesting to see the impact in a more generic setting with more diverse scenes (for example LLFF dataset). Moreover, only two methods are compared Nerfbuster and the proposed method (I discard the vanillar nerf and TensoRF since they have not been developed for this scenario and as such are not really interesting for the discussion). I expected more methods since floaters removal is a popular subject (a recent example from the last CVPR conference is {3}). Additionally, most of the results are shown qualitatively, sometimes with absolutely no comparison whatsoever (Fig. 7). The ablation study is also close to non-existent (literally two visual results comparing a single part of the proposed method). A quantitative study is absolutely necessary. I think a computational cost analysis is also important since adding a per ray transformer can potentially be quite computationally costly.

The presentation could also be improved. The authors rely heavily on sub-indices, which can make it more difficult to read equations (especially when there are too many of them like in, for example, Eq. 1). Abbreviations are also not always defined (for example MAE).

{1} Niemeyer, M., Barron, J. T., Mildenhall, B., Sajjadi, M. S., Geiger, A., & Radwan, N. (2022). Regnerf: Regularizing neural radiance fields for view synthesis from sparse inputs. In Proceedings of the IEEE/CVF Conference on Computer Vision and Pattern Recognition (pp. 5480-5490).

{2} Ehret, T., Marí, R., & Facciolo, G. (2022). Regularization of NeRFs using differential geometry. arXiv preprint arXiv:2206.14938.

{3} Yang, J., Pavone, M., & Wang, Y. (2023). FreeNeRF: Improving Few-shot Neural Rendering with Free Frequency Regularization. In Proceedings of the IEEE/CVF Conference on Computer Vision and Pattern Recognition (pp. 8254-8263).

**Questions:**

In the limitation section, the authors mention that the method assumes fixed lightning conditions and no semi-transparent objects in the scene. in understand that the fixed lightning conditions is necessary for the appearance decomposition but it is not clear for me why semi-transparent objects would be problematic (especially for the "defogging" part), could the authors comment on that?

See also the different remarks in "Weaknesses"

---

### Official Review · Reviewer_pRKJ · 2023-11-01

**Soundness:** 1 poor
**Presentation:** 3 good
**Contribution:** 2 fair
**Rating:** 3
**Confidence:** 4

**Summary:**

The paper tackles the problem of floating artifacts in Neural Radiance Fields. The key idea is two fold. First the authors propose a Transformer that learns a distribution of densities along rays and use this model to clean the ray densities. Secondly, the authors argue the inaccurate appearance modeling in plain NeRFs yields floating artifacts and they hence propose to decompose the appearance into a view-independente and a view-dependent part with SphericalHarmonics. The method is applied to the Shiny Blender dataset and scenes from the Mipnerf 360 dataset.

**Strengths:**

Contains interesting ideas as to learn the ray distribution.

The Literature overview contains most relevant papers.

**Weaknesses:**

1) The choice of the baseline methods can be improved. Especially to evaluate the appearance decomposition part, it would be good to compare to other existing methods, as an example Ref-NeRF would be a good baseline that contains appearance decomposition. For the larger outdoor scene, MipNerf would be a good baseline.

2) More details on the training, the data and the results of the ray rectification transformer should be provided including results on the ray density profile. I’d suggest adding information about the training data, an example of successful ray clean on ray density plots as well as final rendered images with and without ray cleaning. As a reviewer it is hard to judge the impact of this part with the given information, please provide more evidence.

3) Figure 6 caption does not fit the content.

4) It would be good to have a quantitative evaluation for the Shiny Objects dataset to support the appearance decomposition.

**Questions:**

For cleaning the NeRF scene, I'm wondering if it is necessary to incorporate scene knowledge to the ray rectification. How can the ray rectification transformer clean a density distribution for a scene that it hasn't seen before?